# Thionated levofloxacin derivative: Potential repurposing for cancer treatment and synergism with doxorubicin on doxorubicin-resistant lung cancer cells

Hamza Abumansour[1☯*], Osama H. Abusara[1☯], Mohammad Abu-Sini[1], Wiam Khalil[2], Ali I. M. Ibrahim[1], Amal M. Badawoud[3], Majed S. Al Yami[4,5,6], Dina H. Abulebdah[1], Shiraz Halloush[7]

1 Department of Pharmacy, Faculty of Pharmacy, Al-Zaytoonah University of Jordan, Amman, Jordan,
2 Department of Pharmacology, School of Medicine, The University of Jordan, Amman, Jordan,
3 Department of Pharmacy Practice, College of Pharmacy, Princess Nourah bint Abdulrahman University, Riyadh, Saudi Arabia, 4 Department of Pharmacy Practice, College of Pharmacy, King Saud bin Abdulaziz University for Health Sciences, Riyadh, Saudi Arabia, 5 King Abdulaziz Medical City, National Guard Health Affairs, Riyadh, Saudi Arabia, 6 King Abdullah International Medical Research Center, Riyadh, Saudi Arabia, 7 Department of Clinical Pharmacy and Therapeutics, Faculty of Pharmacy, Applied Science Private University, Amman, Jordan

☯ These authors contributed equally to this work.
* h.abumansour@zuj.edu.jo

## Abstract

### Objectives

Fluoroquinolones, such as levofloxacin (LVX), are extended-spectrum drugs used for the treatment of bacterial infections. Several fluoroquinolone derivatives have shown promising antibacterial and anticancer activities. Our group has earlier synthesized and investigated thionated LVX analogs, compounds **2** and **3**, on A549 (non-small cell lung cancer) cell line and showed promising anticancer activity. The mechanism of cytotoxicity may be, in part, via aldehyde dehydrogenase enzyme inhibition and antioxidation. In this study, compounds **2** and **3** were evaluated on prostate (PC-3), breast (MCF7), colorectal (Caco-2), and small cell lung cancer (H69 and H69AR) cell lines.

### Methods

The anticancer activity was measured using resazurin colorimetric method. Combination treatments with doxorubicin (DOX) were also employed and combination index (CI) value were calculated.

### Results

Compound **3** possessed higher anticancer activity compared to compound **2** on the tested cancer cell lines. Compound **3** had the highest activity on PC-3 cells with $IC_{50}$ value of 3.58 µM. DOX was also tested for comparison and had $IC_{50}$ value of less

**Data availability statement:** The datasets generated during and/or analyzed during the current study are presented within the manuscript.

**Funding:** This work was supported by Al-Zaytoonah University of Jordan grants (30/06/2024-2025 to HA and 27/06/2024-2025 to OHA) and Princess Nourah bint Abdulrahman University Researchers Supporting Project number (PNURSP2025R418 to AMB), Princess Nourah bint Abdulrahman University, Riyadh, Saudi Arabia. The funders had no role in study design, data collection and analysis, decision to publish, or preparation of the manuscript.

**Competing interests:** The authors declare that they have no conflict of interest to declare.

**Abbreviations:** ALDH: aldehyde dehydrogenase; CI: combination index; Cyclic AMP: cyclic adenosine monophosphate; DOX: doxorubicin; FLZ: fluconazole; LVX: levofloxacin; MMP-9: matrix metalloproteinase-9; nTPM: normalized transcript per million; PMA: phorbol 12-myristate 13-acetate; SCLC: small cell lung cancer; TGF-β: transforming growth factor beta.

than 0.5 µM in all tested cell lines except for H69AR (DOX-resistant form of H69), which was 4.62 µM. Combination treatment with DOX resulted in significant reduction of cell viability in PC-3, H69, and H69AR cells, with those on H69 and H69AR cells resulted in additive (CI = 1.0) and synergistic effects (CI = 0.6), respectively.

## Conclusions

Compound **3**, a thionated LVX derivative, showed a promising anticancer activity, prompting its potential repurposing for cancer treatment as well as combination treatment with DOX on DOX-resistant cancer cells.

## 1. Introduction

Fluoroquinolones are a class of broad-spectrum antibiotics commonly used to treat a variety of bacterial infections. They exert their antibacterial activity by inhibiting bacterial DNA gyrase, topoisomerase II, and topoisomerase IV enzymes, which are essential for bacterial nucleic acids functions and repair processes, and thus for proliferation [1,2]. Among others, the commonly used fluoroquinolones clinically are levofloxacin (LVX), ciprofloxacin, and moxifloxacin [2]. They are extended-spectrum antibiotic class that act against infections caused by gram-positive and gram-negative bacteria, including urinary tract infections [3], respiratory tract infection [4], bone infections [5], sexually transmitted diseases [6], and skin infections [7].

Fluoroquinolones have also been investigated for other biological activities apart from antibacterial effects. For example, Huang *et al.* showed that a group of fluoroquinolones including ciprofloxacin, LVX, clinafloxacin, gatifloxacin, and enrofloxacin, effectively suppressed TGF-β and PMA-induced MMP-9 levels and activity in HepG2 and A549 cancer cell cultures in a concentration and time-dependent manner [8]. Furthermore, these fluoroquinolones inhibited TGF-β and PMA-induced cell migration by targeting the p38 and cyclic AMP signaling pathways [8].

Newer fluoroquinolones derivatives have been synthesized and investigated as a potential way to improve their efficacy, their antimicrobial spectrum, and for cancer treatment repurposing efforts via introducing several modifications on the quinolone nucleus. For example, Khan *et al.*, showed that chloroquine, an example of drug with a quinolone nucleus, entrapped in phosphatidylserine liposomes have increased activity against *Cryptococcus neoformans* fungal infections both in *in vitro* and *in vivo* studies as a single treatment or in combination with fluconazole (FLZ), a known antifungal drug [9]. Other examples are thionated LVX derivatives, such as compounds **2** and **3** ([Fig 1]) synthesized by our group, that have shown promising activity against bacteria as well as cancer cells, *in vitro* [10,11].

Other derivatives have been investigated for their mechanism of cytotoxicity and antiproliferative effects. These include arresting the cell cycle and interfering with its phases, inducing cell death, suppressing angiogenesis, interfering with cell migration, and adjusting cell signaling pathways [12]. These modified quinolone derivatives that have shown cytotoxic and antiproliferative effects on various cancer cell lines include:

**Fig 1. The structures of thionated levofloxacin derivatives.**

quinolone derivatives containing morpholin alkylamino side chains [13], 5-chloroquinolin-8-ol derivatives [14], and LVX carboxamides derivatives in which 3-chloro or 4-fluoro substituent on the S-benzyl moiety had positive antiproliferative effect compared to doxorubicin [15].

Recently, a previous study by our group [11] examined the cytotoxicity and the potential mechanism of a thionated LVX derivative (compound **3**) on A549 cell line. It showed a promising anticancer activity with a suggested mechanism via aldehyde dehydrogenase (ALDH) enzyme inhibition [11]. Additionally, the anticancer effect of compound **3** on A549 cell line has been investigated when it has been combined with doxorubicin (DOX) and showed enhanced activity [11].

In this study, we aimed to further investigate the anticancer chemotherapeutic spectra of compounds **2** and **3**. Here in, several cancer cell lines were used; prostate cancer cell line (PC-3), breast cancer cell line (MCF7), colorectal cancer cell line (Caco-2), and small cell lung cancer (SCLC) cell lines (H69 and its DOX-resistant form H69AR). As for the anticancer activity, we also investigated the combination of DOX and the possibility of synergism, knowing that several studies investigate combination treatments as a way to induce synergism [11,16–21].

## 2. Materials and methods

### 2.1. Cell culture

DMEM – high glucose, and RPMI-1640 Medium were obtained from Euroclone, Italy. Heat inactivated Fetal Bovine Serum (FBS) was obtained from Capricorn Scientific, Germany. Prostate cancer cell line (PC-3 (ATCC CRL-1435)), breast cancer cell line (MCF7 (ATCC HTB-22)), colon cancer cell line (Caco-2 [Caco2] (ATCC HTB-37)), and SCLC cell lines (H69 (NCI-H69 [H69] (ATCC HTB-119)) and H69AR (ATCC CRL-11351)) were obtained from ATCC. Complete medium was used for cell growth; PC-3, MCF7, and H69 cells in 10% (v/v) FBS/RPMI, H69AR cells 20% (v/v) FBS/RPMI, and Caco-2 cells in 10% (v/v) FBS/DMEM. Cells were incubated at 37 °C. With the exception of the suspended H69 cells, all other cells are adherent, in which media was aspirated, cells washed with PBS, and fresh complete medium was added for their growth. H69 cells were continuously resuspended in fresh medium.

### 2.2. Cell viability assays using resazurin dye method

Resazurin sodium salt and DOX were obtained from Sigma-Aldrich, USA. Our group has synthesized compounds **2** and **3** [10]. The resazurin dye colorimetric method was used to carry out the cell viability assays as described before [11,22].

Briefly, the seeding density for PC-3, MCF7, Caco-2, and H69AR cells in 96-well plates was $1 \times 10^4$ cells/200 µL/well, whereas H69 cells' seeding density was $1 \times 10^4$ cells/180 µL/well. For the adherent cells, 24 h post-seeding, aspiration of the old medium and the addition of 180 µL of fresh medium were performed. The tested compounds (**2**, **3**, and DOX) were initially dissolved in DMSO and diluted with complete medium (of each cell line) forming several stock solutions with concentrations being ranged from 1 mM to 0.03 µM. Then, 20 µL of the compounds' stock solutions was added and treatments were for 96 h. Wells serving as untreated control and for background fluorescence were prepared. Following 96 h incubation, resazurin dye (20 µL; 0.125 mg/mL in PBS) was added onto wells and plates incubated for 4 h. BioTeK SYNERGY HTX plate reader was then used to record the fluorescence readings at 540 nm (excitation) and 620 nm (emission) for percentage viability calculations compared to the control.

### 2.3. Using doxorubicin for the combination treatments

Cell viability assays for the combination treatments were performed as mentioned above and as previously described [11,19]. The final well concentration of DOX used was at or around the $IC_{50}$ value for each cell line. The final well concentrations used for compound **3** ranged from 12.5 to 0.391 µM.

### 2.4. Combination indices for the combination treatments

Cell viability assays were performed as mentioned above and as previously described [11]. A stock solution (10X) for compound **3** as well as for DOX were prepared at their respective $IC_{50}$ concentration using complete medium. The 10X stock solution of compound **3** was mixed with the DOX's 10X stock solution at a 1:1 ratio forming the initial working concentration. The initial working concentration was diluted 7 times at 1:1 ratio using complete medium. For the adherent cells, medium was aspirated and 200 µL of each mixture was added and plates incubated for 96 h. For the suspended cells (H69), the same procedure was applied but with initially preparing 100X stock solution and 20 µL of the diluted mixtures were added to 180 µL of seeded cells. Later, resazurin assay was conducted as mentioned above. The following equation was used to calculate the combination indices [11,20]:

$$Combination\ Index\ (CI) = \frac{D_1}{D_{x1}} + \frac{D_2}{D_{x2}}$$

$D_1$ and $D_2$ are the new $IC_{50}$ value for the compounds in the combination treatment, while $D_{x1}$ and $D_{x2}$ are the original $IC_{50}$ value for the same compounds when used alone.

### 2.5. Statistical analysis

Data are shown as means and standard deviation (SD). The $IC_{50}$ value were calculated using nonlinear regression analysis. The analysis of the combination experiments (compound + DOX) with DOX alone was performed using one-way ANOVA followed by Tukey's multiple comparisons analysis. GraphPad Prism version 9.0 was used to carry out the analysis.

## 3. Results and discussion

### 3.1. Cell viability assays

Experiments were conducted by treating PC-3, MCF7, Caco-2, H69, and H69AR cell lines for 96 h with DOX and compounds **2** and **3**. Cell viability assays are usually conducted for 24, 48, 72, or 96 h depending on the cell line, investigating gene expression, and/or interactions with targets [11,23–25]. Hence, we opted to choose 96 h time-point to evaluate the maximal possible effect of the compounds to get a full overview. DOX was used for its known anticancer efficacy against several types of cancers [26–28] and for comparison purposes. Compounds **2** and **3** were previously investigated for

their anticancer activity on other cancer cell lines [10,11] and were further investigated in this study on a panel of cancer cell lines. The results are presented in Table 1. Our group had previously evaluated the cytotoxicity of LVX, compound 2, and compound 3 on human fibroblasts and found to be noncytotoxic. Hence, their use in cancer cells to investigate their cytotoxicity is worth studying [10].

Table 1 shows that DOX has the highest cytotoxic effect on all tested cancer cell lines, compared to other compounds, due to its effective anticancer activity against several cancers, such prostate, breast, and colorectal cancers [26–28]. Our data (Table 1) shows that DOX's highest activity, among tested cancer cell lines, was recorded against the SCLC cell line H69 with $IC_{50}$ value of 0.09 μM, which is ~50 times more sensitive compared to its resistant form; H69AR [29–31], with $IC_{50}$ value of 4.62 μM

There were several LVX derivatives that have been investigated previously on various cancer cells and showed anti-cancer activity [32]. In addition, our group had investigated compounds 2 and 3 against non-small cell lung cancer cell lines [10,11]. In this study, compound 2 was only active against PC-3 cell line, while compound 3 was only inactive against MCF7 cell line (Table 1). With the exception of results obtained for MCF7 cell line, compound 3 was more cytotoxic than compound 2 on all tested cancer cell lines. PC-3 cell line was the most sensitive for compound 3 with $IC_{50}$ value of 3.58 μM, followed by H69 cell line (14.20 μM), H69AR cell line (35.50 μM), and Caco-2 cell line (52.34 μM) (Table 1). As in the results obtained for DOX, compound 3 was more active against H69 cell line compared to H69AR cell line. On the other hand, compound 2 was inactive against both cell lines (H69 and H69AR).

In our previous work [11], we have shown that compound 3 has the most antioxidant activity compared to LVX and compound 2. We also have shown that compounds 2 and 3 inhibit ALDH activity with compound 3 resulting in a significant inhibition compared to compound 2 [11]. Hence, we have suggested that the possible mechanism of toxicity may be driven by the antioxidant activity and ALDH inhibition, especially for compound 3 [11]. This was further supported by the molecular docking experiments that have shown that compound 3 has a good binding affinity towards ALDH1A3 enzyme [11].

In this study, in terms of thionated LVX derivatives, compound 3 showed higher cytotoxic activity compared to compound 2 on the tested cancer cell lines, especially in PC-3 cell line. PC-3 cell line is known to express ALDH1A3 enzyme and when it is knocked out, cells' proliferation has been significantly reduced [33]. This correlates with our results in which compound 3, having high affinity towards ALDH1A3 [11], resulted in the most cytotoxic effect against PC-3 cell line with $IC_{50}$ value of 3.58 μM, while on other cancer cell lines the value was higher (Table 1). Furthermore, colorectal cancer resistance to medication is driven via ALDH1A3 upregulation [34] and its inhibition might result in cytotoxicity as it is shown for compound 3 on colorectal cancer cell line (Caco-2) in Table 1. This may further support the mechanism of cytotoxicity for these compounds on cancer cells, particularly compound 3, is driven, in part, via ALDH inhibition.

Research articles using SCLC cell lines is very minimal in literature. We could not identify, from research papers, the expression of ALDH enzymes in SCLC and particularly H69 or H69AR cell lines. Hence, further investigation regarding ALDH enzymes expression in SCLC is highly needed in the future.

Table 1. Cytotoxicity ($IC_{50}$ ±SD) of DOX and compounds 2 and 3 on PC-3, MCF7, Caco-2, H69, and H69AR cell lines after treatment for 96 h; n=3.

| Compounds | PC-3 $IC_{50}$±SD (μM) | MCF7 $IC_{50}$±SD (μM) | Caco-2 $IC_{50}$±SD (μM) | H69 $IC_{50}$±SD (μM) | H69AR $IC_{50}$±SD (μM) |
|---|---|---|---|---|---|
| DOX | 0.35±0.03 | 0.45±0.05 | 0.28±0.05 | 0.09±0.02 | 4.62±1.37 |
| Compound 2 | 21.42±2.18 | > 100 | > 100 | > 100 | > 100 |
| Compound 3 | 3.58±0.28 | > 100 | 52.34±6.88 | 14.20±0.48 | 35.50±0.94 |

SD: standard deviation; DOX: doxorubicin.

The Human Protein Atlas website [35], may be used to identify the expression of specific proteins in several cell lines via RNA expression measured in normalized transcript per million (nTPM). Among the tested cell lines in this study, PC-3 cell line expresses the highest level of ALDH1A3 with a value of 565.6 nTPM, followed by Caco-2 cell line (7.1 nTPM), MCF7 cell line (4.8 nTPM), and H69 cell line (0.7 nTPM). However, there were no data available for H69AR cell line. Again, the high expression of ALDH1A3 enzyme in PC-3 cell line [35] along with compound **3** high binding affinity to it [11], further validates that the mechanism of cytotoxicity for compound **3** is driven, in part, via ALDH inhibition.

Our findings suggest the possibility of repurposing thionated LVX derivatives, especially compound **3**, for the treatment of cancers, particularly prostate cancer, due to their promising cytotoxic effects as shown in **Table 1**.

### 3.2. The use of doxorubicin in combination treatment

Compound **3**, being one of the novel thionated LVX derivatives with higher cytotoxic activity compared to compound **2**, was further selected to investigate its cytotoxic activity in combination with DOX on PC-3, MCF7, Caco-2, H69, and H69AR cell lines.

Experiments were conducted by treating PC-3, MCF7, Caco-2, H69, and H69AR cell lines for 96 h. The concentration of compound **3** was ranged from 12.5 to 0.391 µM. As for compound **3** on PC-3 cell line, these concentrations were above and below its respective $IC_{50}$ value on the tested cell lines (**Table 1**). This is to investigate the effect of toxic and non-cytotoxic concentrations of compound **3** when combined with DOX on the tested cell lines. DOX's concentration used was at or around its respective $IC_{50}$ value on PC-3 cell line (**Table 1**). The results are presented in Figs 2–6.

Figs 2–6 show the cytotoxic effect of compound **3** in combination with DOX compared to DOX alone on PC-3, MCF7, Caco-2, H69, and H69 cell lines, respectively, in which the concentration of DOX was at or around its respective $IC_{50}$ value for each cell line (**Table 1**). In all tested cell lines, there was an observed reduction in cell viability when compound **3** was combined with DOX, compared to compound **3** alone, indicating an enhanced cytotoxic effect of the combination treatments. Moreover, a significant reduction of cell viability, compared to DOX alone at or around its respective $IC_{50}$ value on each cell line, was observed only in PC-3, H69, and H69AR cell lines. A significant decrease in cell viability was observed when DOX was combined with 12.5, 6.25, 3.125, and 1.563 µM of compound **3** (**Fig 2**); with 12.5 µM of compound **3** (**Fig 5**); and with all concentrations of compound **3** (**Fig 6**), compared to DOX alone. Although compound **3** was highly cytotoxic against PC-3 cell

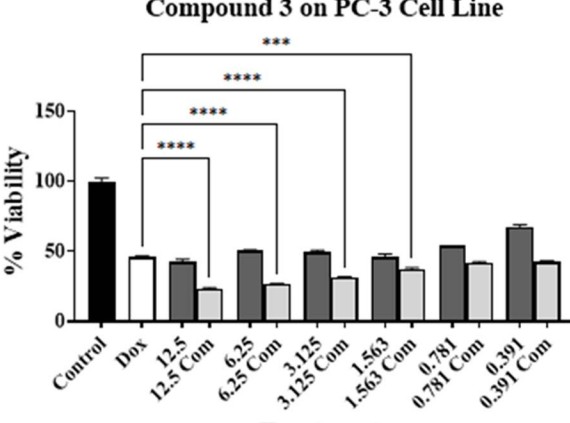

**Fig 2. Cell viability assays using compound 3 on PC-3 cell line alone at various concentrations (µM) (dark grey bars) and in combination with DOX (light grey bars) at or around DOX's $IC_{50}$ value on the cell line.** No treatment is presented with control bar chart (black bar). White bar is DOX at or around its $IC_{50}$ value alone. Experiments were conducted in triplicates at three independent trials with controls (*** $p < 0.001$, and **** $p < 0.0001$).

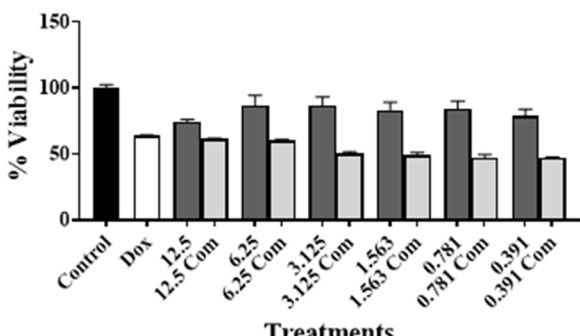

**Fig 3. Cell viability assays using compound 3 on MCF7 cell line alone at various concentrations (µM) (dark grey bars) and in combination with DOX (light grey bars) at or around DOX's IC$_{50}$ value on the cell line.** No treatment is presented with control bar chart (black bar). White bar is DOX at or around its IC$_{50}$ value alone. Experiments were conducted in triplicates at three independent trials with controls.

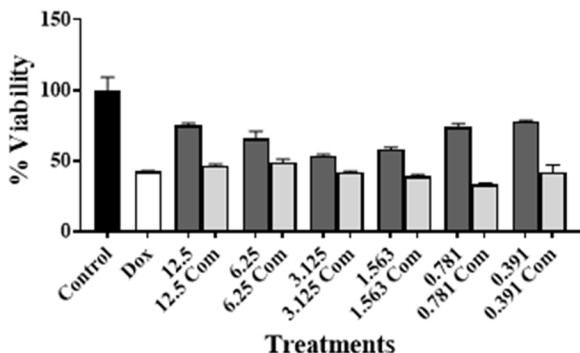

**Fig 4. Cell viability assays using compound 3 on Caco-2 cell line alone at various concentrations (µM)(dark grey bars) and in combination with DO X (light grey bars) at or around DOX's IC$_{50}$ value on the cell line.** No treatment is presented with control bar chart (black bar). White bar is DOX at or around its IC$_{50}$ value alone. Experiments were conducted in triplicates at three independent trials with controls.

line (**Table 1**), synergism with DOX was highly achieved against H69AR cell line at all tested concentrations, while synergism with DOX against PC-3 and H69 cell line was achieved with relatively the higher concentrations of compound **3**.

Interestingly, high synergism was achieved when compound **3** was combined with DOX on H69AR cell line, which is known to be DOX-resistant cell line, as mentioned above. This combination resulted in higher cytotoxic activity in the DOX-resistant cell line compared to the DOX-sensitive cell line (H69) (**Figs 5** and **6**). This might indicate the potential use of compound **3** for the treatment of DOX-resistant cancers in combination treatments with DOX.

Compound **3** might have enhanced DOX cytotoxic activity or vice versa. It has been previously suggested that ALDH-affinic compounds may act as P-glycoprotein substrates, thus allowing more DOX entry into the cell, since DOX is a P-glycoprotein substrate and a competition on the binding site would occur [19]. However, H69AR cells lack P-glycoproteins [31] thus indicating that this suggested mechanism of synergism is invalid, in part, for H69AR cell lines. Furthermore, P-glycoproteins are expressed minimally in PC-3, MCF7, Caco-2, and H69 cell lines as obtained from The Human Atlas Protein website with nTPM

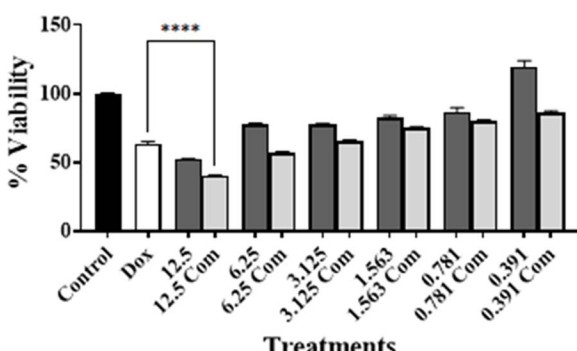

**Fig 5. Cell viability assays using compound 3 on H69 cell line alone at various concentrations (µM) (dark grey bars) and in combination with DOX (light grey bars) at or around DOX's IC$_{50}$ value on the cell line.** No treatment is presented with control bar chart (black bar). White bar is DOX at or around its IC$_{50}$ value alone. Experiments were conducted in triplicates at three independent trials with controls (**** $p < 0.0001$).

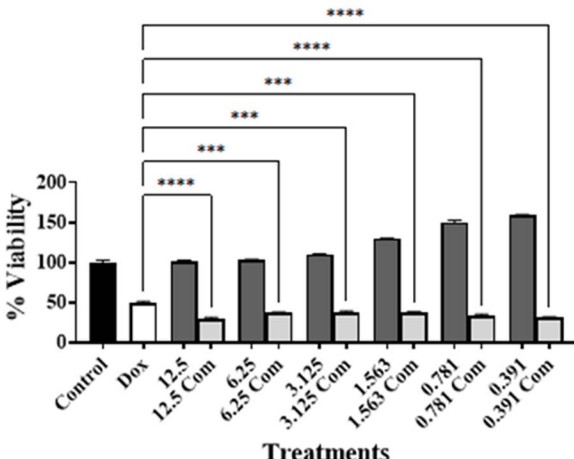

**Fig 6. Cell viability assays using compound 3 on H69AR cell line alone at various concentrations (µM) (dark grey bars) and in combination with DOX (light grey bars) at or around DOX's IC$_{50}$ value on the cell line.** No treatment is presented with control bar chart (black bar). White bar is DOX at or around its IC$_{50}$ value alone. Experiments were conducted in triplicates at three independent trials with controls (*** $p < 0.001$, and **** $p < 0.0001$).

value of 0.0, 0.1, 0.0, and 0.3, respectively [36]. Again, the possibility of compound **3** to act as P-glycoprotein substrate would not be valid to enhance DOX's cytotoxicity or the reason for the enhanced cytotoxicity of the combination treatment.

It might be that DOX that is enhancing compound **3** cytotoxic activity in a mechanism that requires further investigation. Our findings further show the potential use and repurposing of thionated LVX derivatives for the treatment of cancers as well as their combination with DOX may also enhance the cytotoxic effect.

### 3.3. Calculating the combination index for the combination treatment

Significant enhancement of cytotoxicity was observed against PC-3, H69, and H69AR cell lines when compound **3** was combined with DOX (**Figs 2**, **5**, and **6**). The effect of combining compound **3** with DOX on PC-3, H69, and H69AR cell

Table 2. IC$_{50}$ value (µM±SD) for compound 3 as single agent and in combination with DOX and for that of DOX as single agent and in combination with compound 3, along with the calculated combination indices for the combination treatments on PC-3, H69, and H69AR cell lines after treatment for 96h; n=3.

| Treatments | Cell Lines | | |
|---|---|---|---|
| | PC-3 | H69 | H69AR |
| IC$_{50}$ for Compound 3 as Single Agent | 3.58±0.28 | 14.20±0.48 | 35.50±0.94 |
| IC$_{50}$ for Compound 3 in Combination Treatment | 18.15±0.05 | 3.53±0.23 | 9.82±1.16 |
| IC$_{50}$ for the DOX as Single Agent | 0.35±0.03 | 0.09±0.02 | 4.62±1.37 |
| IC$_{50}$ for the DOX in Combination Experiment | 2.11±0.03 | 0.07±0.01 | 1.48±0.16 |
| Combination Index | 11.1* | 1.0** | 0.6*** |

SD: standard deviation; DOX: doxorubicin; *: antagonistic effect; **: additive effect; ***: synergistic effect.

lines was further analyzed via the Combination Index (CI) calculations [37]. The combination treatments may be identified as synergistic (CI<1), additive (CI=1), or antagonistic (CI>1) against the tested cancer cell lines. The experiments would also show the changes of the IC$_{50}$ value for each compound within the combination treatment. The results are presented in **Table 2**.

An antagonistic effect was observed against PC-3 cell line, in which a CI value of 11.1 was achieved. In addition, the IC$_{50}$ value for compound **3** and DOX were higher than those obtained when used alone. Although this may contradict the results obtained for PC-3 cell line as presented in **Fig 2**, it may be that the concentrations used for compound **3** within this experiment have affected the cytotoxic effect of DOX and this would not mean inefficacy in general [11,20].

On the other hand, the results in **Table 2** show that the IC$_{50}$ value for compound **3** and DOX against H69 and H69AR cell lines were lower compared to their value when used alone, indicating an enhancement of cytotoxicity. Consequently, the CI value against H69 cell line was 1.0 indicating an additive effect of the combination treatment and a CI value of 0.6 against H69AR cell line indicating a synergistic effect of the combination. These results can be correlated with the results obtained in **Figs 5** and **6**, which shows a significant reduction in cell viability, especially against H69AR cell line. This further suggests the potential use of compound **3** for the treatment of DOX-resistant cancers.

## 4. Conclusions

In conclusion, we have shown that compound 3, a thionated LVX derivative, has a concentration-dependent anticancer activity against a panel of cancer cell lines, *in vitro*, further indicating the possibility of repurposing them for cancer treatment. Our results here and within our previous study [11] further validate that the mechanism of anticancer activity for the compound **3** may be, in part, via ALDH1A3 inhibition. In addition, combining DOX with compound **3** enhanced the cytotoxic activity against cancer cell lines, namely H69 and its DOX-resultant form H69AR, forming additive and synergistic effects, respectively.

## Acknowledgments

We would like to thank Al-Zaytoonah University of Jordan (Amman, Jordan) and Princess Nourah bint Abdulrahman University (Riyadh, Saudi Arabia).

## Author contributions

**Conceptualization:** Hamza Abumansour, Osama H. Abusara.

**Formal analysis:** Hamza Abumansour, Osama H. Abusara.

**Funding acquisition:** Hamza Abumansour, Osama H. Abusara, Amal M. Badawoud.

**Methodology:** Hamza Abumansour, Osama H. Abusara, Mohammad Abu-Sini, Wiam Khalil, Ali I. M. Ibrahim, Dina H. Abulebdah, Shiraz Halloush.

**Project administration:** Hamza Abumansour, Osama H. Abusara, Amal M. Badawoud, Majed S. Al Yami.

**Supervision:** Hamza Abumansour, Osama H. Abusara, Amal M. Badawoud, Majed S. Al Yami.

**Writing – original draft:** Hamza Abumansour, Osama H. Abusara.

**Writing – review & editing:** Hamza Abumansour, Osama H. Abusara, Wiam Khalil.

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
