## [Decision Letter · Decision Letter 0]

17 Apr 2025

Dear Dr. Abusara,

Thank you for submitting your manuscript to PLOS ONE. After careful consideration, we feel that it has merit but does not fully meet PLOS ONE’s publication criteria as it currently stands. Therefore, we invite you to submit a revised version of the manuscript that addresses the points raised during the review process.

We look forward to receiving your revised manuscript.

Kind regards,

Ruo Wang

Academic Editor

PLOS ONE

“This work was supported by Al-Zaytoonah University of Jordan grants (30/06/2024-2025 and 27/06/2024-2025). Princess Nourah bint Abdulrahman University Researchers Supporting Project number (PNURSP2025R418), Princess Nourah bint Abdulrahman University, Riyadh, Saudi Arabia.”

“This work was supported by Al-Zaytoonah University of Jordan grants (30/06/2024-2025 and 27/06/2024-2025). Princess Nourah bint Abdulrahman University Researchers Supporting Project number (PNURSP2025R418), Princess Nourah bint Abdulrahman University, Riyadh, Saudi Arabia.”

“This work was supported by Al-Zaytoonah University of Jordan grants (30/06/2024-2025 and 27/06/2024-2025). Princess Nourah bint Abdulrahman University Researchers Supporting Project number (PNURSP2025R418), Princess Nourah bint Abdulrahman University, Riyadh, Saudi Arabia.”

Reviewers' comments:

Reviewer's Responses to Questions

**Comments to the Author**

1. Is the manuscript technically sound, and do the data support the conclusions?

Reviewer #1: Yes

Reviewer #2: Yes

Reviewer #3: Partly

Reviewer #4: Yes

2. Has the statistical analysis been performed appropriately and rigorously?

Reviewer #1: Yes

Reviewer #2: Yes

Reviewer #3: Yes

Reviewer #4: Yes

3. Have the authors made all data underlying the findings in their manuscript fully available?

Reviewer #1: Yes

Reviewer #2: Yes

Reviewer #3: Yes

Reviewer #4: Yes

4. Is the manuscript presented in an intelligible fashion and written in standard English?

Reviewer #1: Yes

Reviewer #2: Yes

Reviewer #3: Yes

Reviewer #4: Yes

Reviewer #1: Two thionated Levofloxacin derivatives 2 and 3 were assayed for cytotoxicity in vitro on the prostate (PC-3), breast (MCF7), colorectal (Caco-2), and small cell lung cancer (H69 and H69AR) cell lines using resazurin colorimetric method. Combination treatments with doxorubicin were also employed. The group has previously examined the cytotoxicity and the potential mechanism of a thionated LVX derivative 3 on A549 cell line. Based on the previous study on the mechanism of compound 3, the group assumed that the mode of anticancer activity of two compounds involves inhibiting aldehyde dehydrogenase. This study is in my view additive. There is nothing novel, but just varying types of cancer cell lines with no novelty.

Reviewer #2: PONE-D-25-14330 Section / Question

Title & Abstract

- Does the title and abstract effectively capture the content and focus of the manuscript? YES

- Suggestions for improvement (if any):

1. Of Title; Please remove ‘’s’’ from ‘’derivatives’’ (supposedly only one derivative (i.e. compound 3) was found efficacious, unlike compound 2.

Of Abstract;

2. Please mind that both action mechanisms of cytotoxicity were SPECULATIVELY suggested based to previous report generated by the investigators using A549 lung cancer cell line. Neither molecular mechanism was investigated in the CURRENT Report's CELL LINES (PC3, MCF7, CACO2 and H69 with its resistant H69R!); otherwise the manuscript would have been enriched in depth!

3. Levofloxacin’s cytotoxicities in respective cell lines (PC3, MCF7, CACO2 and H69 with its resistant H69R!) could have provided an in depth comparisons for examined derivatives.

4. Studies of SELECTIVE cytotoxicities of all test compounds (namely doxorubicin, Levofloxacin, both compounds 2 and 3) in normal cells would have proved maximally supportive of their significant chemotherapeutic efficacies.

5. Please use ‘’value’’ following ‘’IC50’’ every time it is mentioned in Tables and draft’s text

6. Please mind suggested changes and alterations included within submission draft (as comments within side bubbles!)

7. Inclusion of Doxorubicin IC50 values within abstract (for substantial discrepancies of efficacy/potency/safety from test derivatives) can enhance substantially the study NOVEL outcomes of NOVEL LVX derivatives.

Introduction

- Is the background and information provided adequate for understanding the research? YES

- Are there additional aspects to incorporate? No!

all satisfactory inclusions are elaborated.

Material and Methods

- Are the methods described with sufficient clarity? YES

- Is there enough detail to enable replication of experiments? -YES

- Are statistical methods appropriate and aligned with objectives? -YES

- Concerns or considerations regarding the methods (if any):

Kindly see above comments in ‘’ABSTRACT’’ section; as there are key points needing further elaboration / justifications (very minor in total) that can signify the study’s Novel outcomes maximally!

Results

- Are the results novel and meaningful?

Yes

- Does the study make a meaningful contribution to the field?

Yes

- Are the results plausible and credible?

Yes

Discussion

- Do the discussion correlate with the results?

Yes

- Are findings relevant to the objectives and broader research context?

Yes

Conclusion

- Do the conclusions align with the findings?

Yes

Figures & Tables

- Are figures and tables clear, legible, and free of unnecessary modifications? YES

Others

- Are the references relevant? YES

- Are the references in the correct style? Mostly YES!

Please ADD DOIs of cited references, in accordance with PLOS ONE stylistic guidelines and format of cited literature

Recommendations to Editor

- Recommendation:

o Revise (minor)

- Would you review a revision? Yes

Additional Comments

_Based to Abu Mansour et al., (2024) publication; this manuscript can be a significantly added value for antiproliferation efficacies of LVX derivatives.

Hence Minor elaborations in experimental design and results (already based to a couple of tables mainly and a single set of figures) can maximally enhance chances of study impact, coherence and integrity!

Reviewer #3: The manuscript describes the anticancer potential of two thionylated derivatives of levofloxacin. The presented data may be of interest to readers. Overall, the manuscript is well-written and clear. I have some suggestions for the authors:

The discussion on the mechanism of action is highly speculative. I suggest at least demonstrating that the proposed target protein (ALDH1A3) is present in the tested cell lines or performing an assay with the recombinant protein.

Please present standard deviation (SD) instead of the standard error of the mean (SEM).

Use higher-quality images for Figures 1 and 2.

Reviewer #4: Dear Authors,

The manuscript is technically sound and the data support the conclusions. The statitical analysis have done appropriately and rigurously. All the data shown in the manuscript was explained appropiately. The presentation of information is well organized and with a good english. However, I have a something to ask you: why the cell viability assays were determined at 96 h? it is common to do such experiments at 24 or 48 hr. So I will suggest you to explain the reason why the cell viability was determined at 96 h.

**Do you want your identity to be public for this peer review?** For information about this choice, including consent withdrawal, please see our Privacy Policy

Reviewer #1: No

Reviewer #2: **Yes: ** VIOLET KASABRI/PROFESSOR OF BIOMEDICINE & LIFE SCIENCES

Reviewer #3: No

Reviewer #4: No

---

## [Author Response · Author response to Decision Letter 1]

19 Apr 2025

Response to Academic Editor and Reviewers

Cover Letter

Dear Editor-in-Chief,

We would like to thank you for giving us the opportunity for our manuscript “PONE-D-25-14330”, which is entitled “Thionated Levofloxacin Derivatives: Potential Repurposing for Cancer Treatment and Synergism with Doxorubicin on Doxorubicin-Resistant Lung Cancer Cells” to be reviewed for possible publication at your respected journal. We also extend our thanks for the respected reviewers for their time in reviewing the manuscript along with their comments and suggested changes that have enhanced the manuscript.

Please find below our responses for the academic editor (regarding journal requirements) as well as a point-by-point response for the reviewers’ comments. Our responses are highlighted in red in the uploaded "Response to Reviewers" file as well as in the “Revised Manuscript with Track Changes” file.

Sincerely,

Osama Haitham Abusara, PhD

Submitting Author

A) Journal requirements:

These style requirements have been made to the revised file.

“This work was supported by Al-Zaytoonah University of Jordan grants (30/06/2024-2025 and 27/06/2024-2025). Princess Nourah bint Abdulrahman University Researchers Supporting Project number (PNURSP2025R418), Princess Nourah bint Abdulrahman University, Riyadh, Saudi Arabia.”

Please note that the Role of Funder statement reads as follows:

“This work was supported by Al-Zaytoonah University of Jordan grants (30/06/2024-2025 and 27/06/2024-2025). Princess Nourah bint Abdulrahman University Researchers Supporting Project number (PNURSP2025R418), Princess Nourah bint Abdulrahman University, Riyadh, Saudi Arabia.”

“This work was supported by Al-Zaytoonah University of Jordan grants (30/06/2024-2025 and 27/06/2024-2025). Princess Nourah bint Abdulrahman University Researchers Supporting Project number (PNURSP2025R418), Princess Nourah bint Abdulrahman University, Riyadh, Saudi Arabia.”

Funding-related text has been removed from the manuscript as requested. The Acknowledgment section in the manuscript has been updated as follows:

“We would like to thank Al-Zaytoonah University of Jordan (Amman, Jordan) and Princess Nourah bint Abdulrahman University (Riyadh, Saudi Arabia).”

Also, please note the amended Funding Statement for the project, which reads as follows:

“This work was supported by Al-Zaytoonah University of Jordan (Amman, Jordan) with grants number: 30/06/2024-2025 and 27/06/2024-2025. The work was also supported by Princess Nourah bint Abdulrahman University (Riyadh, Saudi Arabia) via Princess Nourah bint Abdulrahman University Researchers Supporting Project with project number: PNURSP2025R418.”

Captions have been added to the manuscript for each figure.

B) Reviewers' comments:

Reviewer #1: Two thionated Levofloxacin derivatives 2 and 3 were assayed for cytotoxicity in vitro on the prostate (PC-3), breast (MCF7), colorectal (Caco-2), and small cell lung cancer (H69 and H69AR) cell lines using resazurin colorimetric method. Combination treatments with doxorubicin were also employed. The group has previously examined the cytotoxicity and the potential mechanism of a thionated LVX derivative 3 on A549 cell line. Based on the previous study on the mechanism of compound 3, the group assumed that the mode of anticancer activity of two compounds involves inhibiting aldehyde dehydrogenase. This study is in my view additive. There is nothing novel, but just varying types of cancer cell lines with no novelty.

We thank the respected reviewer for the comments.

Reviewer #2: PONE-D-25-14330 Section / Question

We thank the respected reviewer for the comments. Please find below our responses.

Title & Abstract

- Does the title and abstract effectively capture the content and focus of the manuscript? YES

- Suggestions for improvement (if any):

1. Of Title; Please remove ‘’s’’ from ‘’derivatives’’ (supposedly only one derivative (i.e. compound 3) was found efficacious, unlike compound 2.

The letter “s” has been removed from the word “derivatives” in the title.

Of Abstract;

2. Please mind that both action mechanisms of cytotoxicity were SPECULATIVELY suggested based to previous report generated by the investigators using A549 lung cancer cell line. Neither molecular mechanism was investigated in the CURRENT Report's CELL LINES (PC3, MCF7, CACO2 and H69 with its resistant H69R!); otherwise the manuscript would have been enriched in depth!

We thank the respected reviewer for this comment. In the future, we are planning to synthesize further levofloxacin derivatives based on Compound 3 to further investigate their cytotoxicity against cancer along with further molecular experiments to validate our hypothesized mechanism of action, which is already supported by docking studies and protein expression from the Human Protein Atlas website.

3. Levofloxacin’s cytotoxicities in respective cell lines (PC3, MCF7, CACO2 and H69 with its resistant H69R!) could have provided an in depth comparisons for examined derivatives.

We thank the respected reviewer for this comment. Indeed, we are planning a different set of experiments involving levofloxacin for cancer treatment in our future work. However, this work was mainly focusing on evaluating the anticancer activity of the thionated levofloxacin derivative (mainly compound 3) and its combination with conventional chemotherapy (doxorubicin).

4. Studies of SELECTIVE cytotoxicities of all test compounds (namely doxorubicin, Levofloxacin, both compounds 2 and 3) in normal cells would have proved maximally supportive of their significant chemotherapeutic efficacies.

Our group had previously evaluated Levofloxacin and compounds 2 and 3 before on human normal cells (dermal fibroblasts) and found to be noncytotoxic even at 200 µM (Ibrahim et al. 2022). As for doxorubicin, it is already a conventional chemotherapy with known efficacy and safety. We have added this information in the Results and Discussion section as follows (Lines 185-188):

“Our group had previously evaluated the cytotoxicity of LVX, compound 2, and compound 3 on human fibroblasts and found to be noncytotoxic. Hence, their use in cancer cells to investigate their cytotoxicity is worth studying [10].”

5. Please use ‘’value’’ following ‘’IC50’’ every time it is mentioned in Tables and draft’s text

Changes have been applied in the manuscript as requested.

6. Please mind suggested changes and alterations included within submission draft (as comments within side bubbles!)

Changes have been applied in the manuscript as requested (changes to the Abstract and adding abbreviations).

7. Inclusion of Doxorubicin IC50 values within abstract (for substantial discrepancies of efficacy/potency/safety from test derivatives) can enhance substantially the study NOVEL outcomes of NOVEL LVX derivatives.

It has been added as requested. The following sentence has been added to the Abstract as follows (Lines 65-66):

“DOX was also tested for comparison and had IC50 value of less than 0.5 µM in all tested cell lines except for H69AR (DOX-resistant form of H69), which was 4.62 µM.”

Introduction

- Is the background and information provided adequate for understanding the research? YES

- Are there additional aspects to incorporate? No!

all satisfactory inclusions are elaborated.

Material and Methods

- Are the methods described with sufficient clarity? YES

- Is there enough detail to enable replication of experiments? -YES

- Are statistical methods appropriate and aligned with objectives? -YES

- Concerns or considerations regarding the methods (if any):

Kindly see above comments in ‘’ABSTRACT’’ section; as there are key points needing further elaboration / justifications (very minor in total) that can signify the study’s Novel outcomes maximally!

Changes to the Abstract has been made.

Results

- Are the results novel and meaningful?

Yes

- Does the study make a meaningful contribution to the field?

Yes

- Are the results plausible and credible?

Yes

Discussion

- Do the discussion correlate with the results?

Yes

- Are findings relevant to the objectives and broader research context?

Yes

Conclusion

- Do the conclusions align with the findings?

Yes

Figures & Tables

- Are figures and tables clear, legible, and free of unnecessary modifications? YES

Others

- Are the references relevant? YES

- Are the references in the correct style? Mostly YES!

Please ADD DOIs of cited references, in accordance with PLOS ONE stylistic guidelines and format of cited literature

DOIs have been added to the cited references. References’ list has been updated according to the journal’s guidelines.

Recommendations to Editor

- Recommendation:

o Revise (minor)

- Would you review a revision? Yes

Additional Comments

_Based to Abu Mansour et al., (2024) publication; this manuscript can be a significantly added value for antiproliferation efficacies of LVX derivatives.

Hence Minor elaborations in experimental design and results (already based to a couple of tables mainly and a single set of figures) can maximally enhance chances of study impact, coherence and integrity!

We thank the respected reviewer for the comments and suggestions. It is highly appreciated.

Reviewer #3: The manuscript describes the anticancer potential of two thionylated derivatives of levofloxacin. The presented data may be of interest to readers. Overall, the manuscript is well-written and clear. I have some suggestions for the authors:

We thank the respected reviewer for the comments. Please find below our responses.

The discussion on the mechanism of action is highly speculative. I suggest at least demonstrating that the proposed target protein (ALDH1A3) is present in the tested cell lines or performing an assay with the recombinant protein.

We have presented in the manuscript the level of RNA expression of the target protein (ALDH1A3) in the tested cell lines from the Human Protein Atlas website. The level of expression correlates with the cell viability results, especially for PC-3 cells, along with docking experiments in our previous work. The following sentences are presented in the manuscript in the Results and Discussion section (Lines 231-238):

“The Human Protein Atlas website [35], may be used to identify the expression of specific proteins in several cell lines via RNA expression measured in normalized transcript per million (nTPM). Among the tested cell lines in this study, PC-3 cell line expresses the highest level of ALDH1A3 with a value of 565.6 nTPM, followed by Caco-2 cell line (7.1 nTPM), MCF7 cell line (4.8 nTPM), and H69 cell line (0.7 nTPM). However, there were no data available for H69AR cell line. Again, the high expression of ALDH1A3 enzyme in PC-3 cell line [35] along with compound 3 high binding affinity to it [11], further validates that the mechanism of cytotoxicity for compound 3 is driven, in part, via ALDH inhibition.”

Please present standard deviation (SD) instead of the standard error of the mean (SEM).

SD has now been used instead of SEM with changes applied into the manuscript.

Use higher-quality images for Figures 1 and 2.

Images have been updated as requested.

Reviewer #4: Dear Authors,

The manuscript is technically sound and the data support the conclusions. The statitical analysis have done appropriately and rigurously. All the data shown in the manuscript was explained appropiately. The presentation of information is well organized and with a good english. However, I have a something to ask you: why the cell viability assays were determined at 96 h? it is common to do such experiments at 24 or 48 hr. So I will suggest you to explain the reason why the cell viability was determined at 96 h.

We thank the respected reviewer for the comments. Indeed, most research articles used 24 or 48 hr to conduct cell viability assays, but we passed through several others that use 72 and 96 hr for several reasons, such as the cell line used, gene expression, and/or target interactions. Hence, we opted to use 96 hr time-point for evaluating the maximal possible effect. We also added the following sentences in the Results and Discussion section (Lines 179-182):

“Cell viability assays are usually conducted for 24, 48, 72, or 96 h depending on the cell line, investigating gene expression, and/or interactions with targets [11, 23-25]. Hence, we opted to choose 96 h time-point to evaluate the maximal possible effect of the compounds to get a full overview.”

---

## [Decision Letter · Decision Letter 1]

4 May 2025

Thionated Levofloxacin Derivative: Potential Repurposing for Cancer Treatment and Synergism with Doxorubicin on Doxorubicin-Resistant Lung Cancer Cells

PONE-D-25-14330R1

Dear Dr. Abusara,

We’re pleased to inform you that your manuscript has been judged scientifically suitable for publication and will be formally accepted for publication once it meets all outstanding technical requirements.

Kind regards,

Ruo Wang

Academic Editor

PLOS ONE

Additional Editor Comments (optional):

Reviewers' comments:

Reviewer's Responses to Questions

**Comments to the Author**

Reviewer #1: All comments have been addressed

Reviewer #4: All comments have been addressed

2. Is the manuscript technically sound, and do the data support the conclusions?

Reviewer #1: Partly

Reviewer #4: Yes

3. Has the statistical analysis been performed appropriately and rigorously?

Reviewer #1: Yes

Reviewer #4: Yes

4. Have the authors made all data underlying the findings in their manuscript fully available?

Reviewer #1: Yes

Reviewer #4: Yes

5. Is the manuscript presented in an intelligible fashion and written in standard English?

Reviewer #1: Yes

Reviewer #4: Yes

Reviewer #1: Cytotoxicity does not define the mode of cancer cell death. Unless the mechanism of cancer cell death has been established, the 'cytotoxicity' should be used in place of 'anticancer'. The authors forwarded excuses on not performing further assays and plan to do that in the near future, hence I previously mentioned that the research constitutes fragmented studies just varying the cell lines (additive))! I am reluctantly abiding by the majority consensus of the reviewers, but still feel strongly that more should have been done for this article to be recommended for publication.

Use 'Fig.' with a dot/full stop as an acronym for figure.

Reviewer #4: Dear Author,

Thank you for made the corrections the manuscript has been improved. The manuscript has good scientific content and contribuite to the knowlege of anticancer agents.

**Do you want your identity to be public for this peer review?** For information about this choice, including consent withdrawal, please see our Privacy Policy

Reviewer #1: No

Reviewer #4: No

---

## [Editor Report · Acceptance letter]

PONE-D-25-14330R1

PLOS ONE

Dear Dr. Abusara,

I'm pleased to inform you that your manuscript has been deemed suitable for publication in PLOS ONE. Congratulations! Your manuscript is now being handed over to our production team.

Kind regards,

on behalf of

Dr. Ruo Wang

Academic Editor

PLOS ONE